# Tree Biomass and Leaf Area Allometric Relations for *Betula pendula* Roth Based on Samplings in the Western Carpathians

**DOI:** 10.3390/plants12081607

**Published:** 2023-04-10

**Authors:** Bohdan Konôpka, Vlastimil Murgaš, Jozef Pajtík, Vladimír Šebeň, Ivan Barka

**Affiliations:** 1National Forest Centre, Forest Research Institute Zvolen, T. G. Masaryka 2175/22, 960 01 Zvolen, Slovakia; bohdan.konopka@nlcsk.org (B.K.); vlastimil.murgas@nlcsk.org (V.M.); jozef.pajtik@nlcsk.org (J.P.); ivan.barka@nlcsk.org (I.B.); 2Faculty of Forestry and Wood Sciences, Czech University of Life Sciences Prague, Kamýcká 129, 165 21 Prague, Czech Republic

**Keywords:** silver birch, young trees, naturally regenerated stands, stem base diameter, tree height, biomass of tree components

## Abstract

Biomass allometric relations are necessary for precise estimations of biomass forest stocks, as well as for the quantification of carbon sequestered by forest cover. Therefore, we attempted to create allometric models of total biomass in young silver birch (Betula pendula Roth) trees and their main components, i.e., leaves, branches, stem under bark, bark, and roots. The models were based on data from 180 sample trees with ages up to 15 years originating from natural regeneration at eight sites in the Western Carpathians (Slovakia). Sample trees represented individuals with stem base diameters (diameter D_0_) from about 4.0 to 113.0 mm and tree heights between 0.4 to 10.7 m. Each tree component was dried to constant mass and weighed. Moreover, subsamples of leaves (15 pieces of each tree) were scanned, dried, and weighed. Thus, we also obtained data for deriving a model expressing total leaf area at the tree level. The allometric models were in the form of regression relations using diameter D_0_ or tree height as predictors. The models, for instance, showed that while the total tree biomass of birches with a D_0_ of 50 mm (and a tree height of 4.06 m) was about 1653 g, the total tree biomass of those with a D_0_ of 100 mm (tree height 6.79 m) reached as much as 8501 g. Modeled total leaf areas for the trees with the above-mentioned dimensions were 2.37 m^2^ and 8.54 m^2^, respectively. The results prove that diameter D_0_ was a better predictor than tree height for both models of tree component biomass and total leaf area. Furthermore, we found that the contribution of individual tree components to total biomass changed with tree size. Specifically, while shares of leaves and roots decreased, those of all other components, especially stems with bark, increased. The derived allometric relations may be implemented for the calculation of biomass stock in birch-dominant or birch-admixed stands in the Western Carpathians or in other European regions, especially where no species- and region-specific models are available.

## 1. Introduction

Silver birch (*Betula pendula* Roth) is a short-lived broadleaved tree species which is distributed throughout most of the European territory, with the highest frequency in the northern countries [1]. As for Slovakia, the latest National Forest Inventory data showed that although birches (mostly *B. pendula* less *B. pubescens* Ehrh.) contributed only by about 1% to the growing stock on forest land, a much larger portion, i.e., about 6.5%, was estimated for nonforest, prevailingly former agricultural lands [2]. The inventory data further suggested that birches are more frequent in young stands than in older ones. This is related to large-scale forest disturbances in Slovakia, especially in the last two decades [3]. The disturbances were followed by the natural forest regeneration with a high share of pioneer tree species, prevailingly birches [4]. Therefore, the contribution of birch to standing stock on both forest and former nonforest land would very likely increase in the near future as a result of its biomass accumulation over time. Moreover, since we can expect further large-scale disturbances in the forests of Slovakia, birches might be important pioneer tree species in forest restoration and would create favorable growth conditions (microclimate and soil properties) for other tree species [5,6]. Their advantage is that in contrast to other pioneer tree species, birches are not an attractive food source for large wild herbivories, such as red deer (*Cervus elaphus* L.). Thus, they are obviously not destroyed by browsing, which is a very important disturbance factor in territories with typically high population densities of this game [7].

In the conditions of Central and Western Europe, birches are nearly irrelevant to the wood processing industry, but they fulfill a variety of ecological roles [5]. The situation is different in Scandinavian and Baltic countries due to the high contribution of birch to forest stock, e.g., in Finland and Norway it accounts for about 16%, in Estonia and Latvia as much as 20%, but also due to the long tradition of birch wood utilization in industries [8,9,10]. The recent energetic crisis in Europe has increased demands for wood as a heating medium and for energy production. Therefore, we assume that birch might also become more important in the European market than it was in the past.

The latest papers from Dubois et al. [11,12] indicated that birch wood might have a high potential for forest-based industry, but management in forestry must change towards the production of large-size logs. In Slovakia, Konôpka et al. [13] showed that silver birch at postdisturbance areas in the High Tatra Mts. overgrew all other tree species and had much higher stem wood production than Norway spruce. Another study [14] manifested that 15-year-old silver birch trees were as high as 12 m, and their diameters at breast height (DBH) exceeded 20 cm. 

Many authors (e.g., [6,15,16,17]) have pointed out the importance of birches from an ecological perspective, especially for the improvement of soil functioning. Birch trees are positively evaluated in terms of direct and indirect contributions to flora and fauna biodiversity [18,19,20]. Birches are one of the suitable choices in enriching species diversity, especially in coniferous monocultures of North American and European countries that were intensively managed [11,21]. At the same time, this is also a way to continuously enhance forest ecosystem services [22,23]. 

As it was mentioned, wood (or biomass) of silver birch was not a part of any emphatic attention of forestry and forest-related sectors in Central Europe; therefore, research activities in this area were perhaps also rather sparse here. Recently, Bronisz and Mehtätalo [24] made mixed-effect biomass models for young birch stands on postagricultural lands in Poland. The authors explained that allometric biomass models express total tree biomass and/or biomass of each tree component as a function of tree-level predictors, such as DBH, D_0_, or tree height (H). Sometimes, a combination of two predictors (DBH and H or D_0_ and H) is even implemented [25]. Other biomass models for birch originate mostly from Scandinavian countries, specifically from Finland [26], Norway [27], and Sweden [28]. Thus, after reviewing the available literature, we can unequivocally conclude that allometric models for silver birch growing under the conditions of the Carpathian Arch are missing. 

The main aim of this work was to construct allometric models for silver birch biomass considering separate tree components (leaves, branches, stem under bark, bark, and roots), as well their aggregated forms, i.e., aboveground and whole-tree biomass. Furthermore, we attempted to make an allometric model for leaf area at the tree level. All models were based either on diameter D_0_ or tree height. Since the sampled trees originated from eight sites all over the Slovak territory, our allometric relations might be usable, especially for the Western Carpathians.

## 2. Results

Our set of sampled trees represented silver birch individuals with diameter D_0_ from approx. 4 mm to 113 mm and heights from 0.4 m to 10.7 m (Table 1). Hence, these ranges of diameters and heights might further cover biomass modeling in the form of allometric relations.

Both derived relations, i.e., between diameter D_0_ and tree height, and especially between diameters D_0_ and DBH, were close (see the values of R^2^ and RMSE in Table 2; Figure 1). Since we did not use DBH as a predictor in biomass models (the main reason was that some trees were shorter than 130 cm), the relationship between D_0_ and DBH can also be implemented by potential users to calculate biomass using DBH.

The biomass models of specific components showed that diameter D_0_ was a better predictor than tree height (compare values of R^2^ as well as RMSE in Table 3; Figure 2 and Figure 3). The exception was found in relations for stem under bark biomass and bark biomass. These were well expressed by both predictors. At the same time, models of aboveground biomass as well as total tree biomass with diameter D_0_ were more precise than those with tree height (Table 4). The model, for instance, showed that while the total tree biomass of birches with a D_0_ of 50 mm (tree height 4.06 m) was about 1653 g, that of those with a D_0_ of 100 mm (tree height 6.79 m) reached as much as 8501 g. The RMSEs obtained for the estimated total aboveground and total biomass with diameter D_0_ as predictor variable were 527.66 g and 618.76 g, respectively. Slightly higher values of RMSE were obtained for estimated total aboveground (RMSE = 527.82 g) and total biomass (RMSE = 618.96 g), calculated as the sum of the individual tree components. The RMSEs obtained for the estimated total aboveground and total biomass with tree height H as predictor variable were 1027.36 g and 1249.33 g, respectively. RMSE values for the estimated total aboveground and total biomass calculated as the sum of the individual tree components were 1037.94 g and 1260.11 g, respectively. 

The calculated contributions of tree components to total biomass indicated changes in biomass structure with tree size (Figure 4). While shares of leaves and roots decreased, those of all other components, especially stem with bark, increased with the increasing tree size. The results suggest that the smallest trees had the largest share of roots, while the greatest contribution of stem under bark was typical of the biggest trees.

Both diameter D_0_ and tree height were shown to be suitable predictors of stem volume under bark and stem volume over bark (Table 5; Figure 5). For instance, the models showed that while the stem volumes under or over the bark of birches with a diameter D_0_ of 50 mm were about 1377 cm^3^ and 1724 cm^3^, respectively, those with a diameter D_0_ of 100 mm had a stem volume under bark of about 7487 cm^3^ and over bark of about 9684 cm^3^. Since data on volume as well as on biomass of stems were available, we were also able to calculate stem wood density. Our analyses showed that stem wood density did not change with tree size, and there were no significant differences among trees from individual sampled sites. Therefore, we just calculated a mean value of the stem wood density. The values (mean ± standard deviation) were 0.46 ± 0.06 g cm^−3^ and 0.50 ± 0.06 g cm^−3^ for stems under and over bark, respectively. The results indicate a higher density of bark than wood in stems.

As for the total tree leaf area, again, diameter D_0_ was a much better predictor than tree height (Table 6; Figure 6). The model suggested, for instance, that trees with a diameter D_0_ of 50 mm have a total leaf area of about 2.37 m^2^, while those with a diameter D_0_ of 100 mm have as much as a four-fold larger leaf area. Here, we would like to repeat the fact that the leaf area at the tree level was expressed by converting the leaf biomass by means of SLA. The SLA was calculated for every single sampled tree, and our analyses showed that the values of SLA did not change with tree size, and no effect of site was found, either. Therefore, we used the mean value of SLA, which was 147 cm^2^ g^−1^, while the standard deviation was ±87 cm^2^ g^−1^.

An interesting output was obtained from the relation between the tree dimension (diameter D_0_ or tree height) and the leaf area ratio (LAR; i.e., the ratio between the total tree leaf area and the total tree biomass). The values of LAR decreased with the tree size, first sharply then only mildly (Table 6; Figure 7). Here, tree height was a better predictor than diameter D_0_. 

Performance measures of regression models based on k-fold cross-validation method are given in Table A1 (see Appendix A).

## 3. Discussion

Our biomass models for tree components of silver birch suggested that stem diameter D_0_ is a more suitable independent variable (higher values of R^2^ and lower RMSE) than tree height. This finding is in accordance with some previous works (e.g., [29,30,31] that claimed stem diameter D_0_ but more often DBH as the most relevant variables for estimating the biomass of individual tree components as well as of the whole tree. Rather surprisingly, tree height was nearly as precise a predictor as diameter D_0_ in relations for stem volume—both under and over bark. Moreover, in the case of LAR, tree height appeared as a better predictor than diameter D_0_. This is a rather unexpected result since both parts of the indicator, i.e., total tree leaf area (numerator) and total tree biomass (denominator), were better expressed by diameter D_0_ than height.

Here, we would like to introduce the practical aspect of the predictors. In the case of field work in young stands, DBH seems to be the most comfortable measure. However, this tree characteristic cannot be measured on individuals with heights below 130 cm, only practically on trees which are slightly higher than 130 cm. Although tree height was found to be a less precise predictor in derived biomass models, its advantage is that it can be used for any tree, i.e., not only for those with heights over 130 cm. Anyway, our paper provides opportunities to use any predictor, i.e., diameter D_0_ directly or tree height, but also DBH by implementing the derived mathematical relation between the diameters. 

Our relations for tree components with regard to tree dimensions indicated different rates of biomass increase. This was the most evident when comparing leaves and stems. For instance, while leaf biomass in trees with a D_0_ of 20 mm was 24 times lower than in trees with a D_0_ of 100 mm, the difference in stem under bark biomass was as much as 55-fold—but in the opposite course. Therefore, the leaves of trees with a D_0_ of 20 mm contributed to total tree biomass more than those with a D_0_ of 100 mm, and at the same time, the reverse situation occurred for stem biomass contribution. As for branches, bark, and roots, the changes in their biomass contributions to the total biomass with tree size were less pronounced. The same courses of leaf and stem contributions to total biomass with changing tree dimensions were stated by Pajtík et al. [31]. They found such changes not only for young broadleaved species, such as *Acer pseudoplatanus* L., *Carpinus betulus* L., *Fagus sylvatica* L., *Fraxinus excelsior* L., *Populus tremula* L., *Salix caprea* L. *Sorbus aucuparia* L., and *Quercus petraea* Liebl., but also for coniferous ones, specifically *Picea abies* L. Karst., *Pinus sylvestris* L., and *Larix decidua* Mill. Moreover, the authors showed that the increasing contribution of branches to total biomass was typical for all species (our birch models manifested this tendency clearly if D_0_ was used as a predictor), while the contribution of roots was rather stable or diminishing with increasing tree size. 

Our results further show decreasing values of LAR with increasing tree size, while the sharpest decrease was typical for small trees. Leaf area is a tree part active in the absorption of solar energy and CO_2_, producing carbohydrates via photosynthesis, thus governing the production of tree biomass. Another indicator used in physiology is Growth Efficiency (GE), which represents the amount of produced biomass (usually on an annual basis) per leaf area unit [32]. The indicators are mutually linked. If we consider our model expressing LAR against tree size (diameter D_0_ or height), much lower GE in very small trees in comparison with larger ones can be judged. This might be related to the limited growth potential in small trees due to small quantities of cells in vascular cambium of woody parts [33]. Anyway, leaf area is important for successful tree growth and development from an early stage. For instance, Fender et al. [34] showed that leaf area and the processes of leaf area development on beech saplings are key determinants of productivity, but at the same time, they are controlled by different environmental factors.

In fact, the principal scientific interest in the quantification of tree biomass and portions of separate components has been prevailingly focused on two main research areas, specifically tree physiology and forest ecology [33,35]. Since tree biomass is the result of photosynthesis, which is performed by leaves, it is essential to study tree growth regarding at least two principal biomass fragments: woody parts (1), i.e., branches, bark, stem under bark, and roots versus leaves (2). This kind of research might elucidate growth strategies in terms of optimizing carbohydrate allocation under certain growth conditions [36,37]. The growth of particular tree components (i.e., biomass allocation) is ruled by a variety of internal (e.g., genetic properties and health status; [38]) and external factors (especially climate, light, soil, and stand conditions; e.g., [39,40]. As for the external factors, Konôpka et al. [15] showed that the intraspecific crown competition in a young silver birch stand, i.e., contrasting light and space conditions, modified not only proportions of leaves, branches, and stem to aboveground biomass but also leaf traits (weight, area, and SLA). Therefore, one idea for future research in this field might be a construction of allometric models that would also consider the competition status. Very likely, the competition stress might be expressed via the bio-sociological position of an individual tree or a stand density (number of trees per unit area with regard to the growth stage).

Finally, it is necessary to point out the need to construct biomass models for young trees, as those established for older trees are absolutely inapplicable due to different biomass allocation patterns at different growth stages [41]. Biomass models in tree compartments in stands of all growth stages/age classes are important, especially if one aims to understand the dynamics of biomass allocation and hence the carbon storage and cycling in forests. In the very initial growth stages, most young trees will not survive due to competitive pressure [42]. Consecutively, their decomposing components contribute to carbon cycling in forest ecosystems. Another need for the quantification of biomass in small trees is related to the recent attempt in most European countries to manage forests according to “close-to-nature” principles [43]. Forest stands under such a regime contain trees of a variety of ages in one place, i.e., small trees, too. Moreover, under the conditions in Europe, close-to-nature management would bring a higher share of species which were traditionally not considered as commercial. It also means soft broadleaved species including birches [20]. 

Besides the many positive impacts of birches on forest ecosystems, their prospect under the ongoing climate change is rather questionable. Although silver birch are generally considered as tolerant to unfavorable climatic and soil conditions (e.g., [5]), some studies indicated their sensitivity to drought stress [44]. Actually, the extreme drought during 2022 in Slovakia caused a very severe discoloration and defoliation of birch trees already in the middle of the growing season. In fact, these symptoms were manifested within birch crowns more intensively than in the rest of the broadleaved tree species (data not published). Therefore, further research on birches should, in addition to other aspects, include issues related to the resistance of birches to harmful agents induced or stimulated by the ongoing climate change.

## 4. Materials and Methods

### 4.1. Stand Selection and Tree Sampling

Silver birch occurs in Slovakia from very low altitudes to approximately 1200 m above sea level [45]. This was the limiting range for searching forest stands suitable for our tree sampling. The preliminary selection of stands was performed using the current national database of forest stands based on the data from Forest Management Plans (see also http://gis.nlcsk.org/lgis/, accessed on 12 January 2023). At the same time, the stands had to grow at moderately fertile sites. In fact, most of them were found on mesotrophic Cambisols. 

The main criteria for the selection of sampling stands were that stands should originate exclusively from natural regeneration, the share of silver birch in tree species composition should be at least 50%, and mean stand age should be below 15 years. Then, we performed field surveys and finally selected eight stands for further tree sampling (Table 7; Figure 8). 

Mean ages of the target forest stands were from 2 to 15 years. Most of the stands were situated on the edges of forest complexes and had rather sparse canopies, with the dominant position of silver birch in the main canopy layer. No silvicultural (precommercial) cuttings were performed in any sampled birch stands.

Within the eight selected forest stands, a total of 180 individuals of silver birch were subjected to our sampling and measurements. Specifically, at each site, between 20–25 birch trees were chosen to cover the entire height range of each stand. Before cutting a tree, both D_0_ and DBH diameters were measured with a digital caliper with a precision of 0.1 mm in two perpendicular directions. The sampling tree was cut at the ground level and its height was measured with metal tape with a precision of 1 cm. Then, 15 leaves were randomly selected and cut along the vertical profile of the tree crown and inserted in a paper envelope marked with specific codes (i.e., site name and tree number). The aboveground parts of each tree were separated into branches with leaves and a stem. The tree components were packed in paper bags marked with specific codes. Moreover, underground tree parts (root system) were carefully excavated following a principle to include all root segments exceeding thicknesses of 2 mm. The samples of underground parts were packed in paper bags, marked with specific codes, and transported together with all other samples to a laboratory. There, leaves were separated from branches and packed. 

The stems were divided into approx. 50 cm-long sections (the smallest trees into shorter sections following the principle to have at least three sections per tree). Then, diameters at both ends and in the middle of each section were measured with a digital caliper with a precision of 0.1 mm, always in two perpendicular directions. Bark was peeled off from every stem section and diameters were measured in the same way as before peeling. The subsamples of leaves were first scanned and then dried in an oven under 95 °C for 24 h. Subsequently, each individual leaf was weighed using a precise laboratory microscale (±0.0001 g). Scanned images of leaves were used to estimate their areas with the Easy Leaf Area program [46].

The samples were stored in a dry and well-ventilated room for approximately one month. Afterwards, each tree component, i.e., leaves, branches, stem under bark, bark, and roots were oven-dried under 105 °C until a constant weight was reached. The dried tree components were weighed with a digital scale (±0.1 g). 

### 4.2. Tree Biomass, Leaf Area, and Stem Volume Calculations

First, the relationship between diameter D_0_ and tree height, including statistical characteristics, was described with the following equation:(1)$$H=\frac{D_{0}^{2}}{b_{0}+b_{1}D_{0}+b_{2}D_{0}^{2}}$$
where:H is tree height (m);D_0_ is diameter at stem base (mm);b_0_, b_1_, and b_2_ are parameters to be estimated.

The relationship between measured diameters DBH and D_0_ was described with the following linear function:(2)$$DBH=b_{0}+b_{1}D_{0}$$
where:DBH is diameter at breast height (mm);D_0_ is diameter at stem base (mm);b_0_ and b_1_ are parameters to be estimated.

Then, biomass models for tree components, i.e., roots, stem, branches, and leaves separately, as well as for the whole aboveground and total tree biomass, were derived. We used two approaches for expressing biomass based either on diameter D_0_ (see Equation (3)) or height (Equation (4)) (in g), which are as follows: (3)$$B_{i}=b_{0}D_{0}^{b_{1}}$$
(4)$$B_{i}=b_{0}H^{b_{1}}$$
where:B is the dry biomass of a tree component i (leaves, branches, stem under bark, bark, roots, aboveground biomass or total tree biomass in grams);D_0_ is diameter at stem base (mm);H is tree height (m);b_0_ and b_1_ are parameters to be estimated.

In addition to developed biomass models for tree components, the aboveground biomass and the total tree biomass were also obtained by summing the model predictions of tree components to explore the consistency of biomass additivity.

The biomass distribution into tree parts (i.e., leaves, branches, stem under bark, bark, and roots) with regard to diameter D_0_ and height H is shown using fitted values from derived biomass models.

For the calculation of volume of each log section, we used Newton’s formula:(5)$$V=\frac{L\left(A_{b}+4A_{m}+A_{s}\right)}{6}$$
where:V is the volume of the stem log section (cm^3^);L is the section length (cm);A_b_ is the cross-sectional area at the wider end of the section (cm^2^);A_m_ is the cross-sectional area in the middle of the section (cm^2^);A_s_ is the cross-sectional area at the thinner end of the section (cm^2^).

The total stem volume was calculated as the sum of the volumes (both over and under bark) of all sections. The calculated values of volume and biomass were used to calculate the basic wood density of stems (under and over bark). In general, basic wood density is defined as the dry mass for a given volume (measured in fresh status) wood.

Regression models using a power function (see Equation (3) or Equation (4)) were developed for predicting individual leaf weight (w_f_), leaf area (LA), and specific leaf area (SLA). SLA was calculated from the measured data of leaf weight (w_f_) and leaf area (LA) as follows:(6)$$SLA=\frac{LA}{w_{f}}$$

The LA at the tree level (in m^2^) was calculated by multiplying the mean value of SLA per sampled tree with its dry leaf biomass. Subsequently, we calculated the leaf area ratio (LAR) as a ratio between the total tree leaf area (LA) and the total tree biomass. 

The 95% confidence and prediction intervals for the fitted allometric models were computed by the predFit function in the investr package version 1.4.2 [47].

### 4.3. Model Assessment and Validation

In this study, regression models were fitted to the entire data set, while model validation was accomplished by implementing a variation of the *k*-fold cross-validation method. The data were first partitioned into *k* segments or folds based on sampling site. Subsequently, *k* iterations of training and validation were performed, such that within each iteration a different fold of the data was held out for validation, while the remaining *k* − 1 folds were used for training. The total number of folds was 8.

For model assessment, we computed R-squared (R^2^), the root mean squared error (RMSE), the Akaike information criterion (AIC), and the Bayesian information criterion (BIC). Model performance was evaluated by five model validation statistics on the testing set: R-squared (R^2^), mean absolute error (MAE), mean square error (MSE), root mean square error (RMSE), and relative root mean square error (RMSE%), and then weighted averages were calculated:(7)$$R^{2}=1-\frac{SSE}{SST}=1-\frac{\sum {\left(y_{i}-{\hat{y}}_{i}\right)}^{2}}{\sum {\left(y_{i}-\overset{-}{y}\right)}^{2}}$$
(8)$$MAE=\frac{1}{n}\sum_{i=1}^{n}\left|y_{i}-{\hat{y}}_{i}\right|$$
(9)$$MSE=\frac{1}{n}\sum_{i=1}^{n}{\left(y_{i}-{\hat{y}}_{i}\right)}^{2}$$
(10)$$RMSE=\sqrt{MSE}=\sqrt{\frac{1}{n}\sum_{i=1}^{n}{\left(y_{i}-{\hat{y}}_{i}\right)}^{2}}$$
(11)$$RMSE\%=\frac{RMSE}{\overset{-}{y}}100$$
where:SSE—sum of squares error;SST—sum of squares total;$y_{i}$—observed value;$\hat{y}$—predicted value of *y*;$\overset{-}{y}$—mean value of *y*.

To evaluate additivity of tree biomass components, we calculated the RMSE when estimating the total aboveground and total biomass using the single model in comparison with the RMSE calculated as the sum of the individual tree components.

Statistical analyses were performed using R programming language 4.2.2 [48] and visualized using ggplot2 package version 3.4.1 [49].

## 5. Conclusions

In addition to other scientific benefits, allometric models can serve as a tool for biomass quantification, not only at the tree level but also at the stand level. Thus, the new allometric models of silver birch can serve the calculation of standing biomass stock (possibly fixed amount of carbon) as well as of biomass production (carbon sequestration) in birch stands or mixed stands with the admixture of this species. These kinds of findings are valuable for regional or country-level estimates on carbon accumulated in forests and its further prediction in forest areas, as well as in former agricultural lands recently covered by tree vegetation.

Finally, we would like to point out that our new allometric models for young birch trees fill the existing gap in this field. Previously, our team constructed allometric relations or biomass expansion and conversion factors (BECF; these convert stem volumes to biomass of any tree component) for these tree species: *Acer pseudoplatanus*, *Carpinus betulus*, *Fagus sylvatica*, *Fraxinus excelsior*, *Picea abies*, *Pinus sylvestris*, *Larix decidua*, *Populus tremula*, *Salix caprea, Sorbus aucuparia*, and *Quercus petraea*. Thus, there are already biomass models available for young trees of twelve autochthonous species of the Western Carpathians. Within the group of the most frequent tree species in this region, only biomass models for young *Abies alba* Mill. are still missing, and this might be a task for further research work.

## Figures and Tables

**Figure 1 plants-12-01607-f001:**
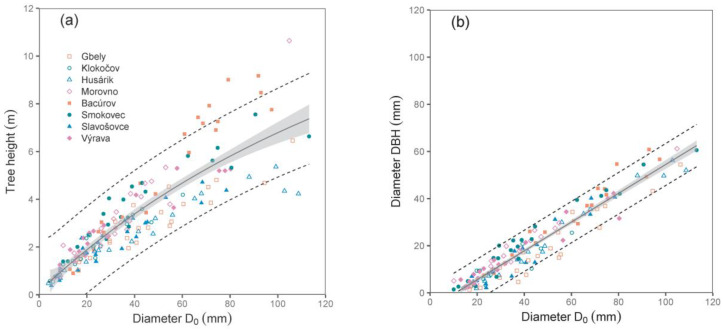
Relations between diameter D_0_ and tree height (**a**), as well as between diameters D_0_ and DBH (**b**) derived from sampled trees of silver birch (characteristics of allometric models are shown in Table 2). Gray areas indicate 95% confidence intervals, and dashed lines denote ranges of 95% prediction intervals.

**Figure 2 plants-12-01607-f002:**
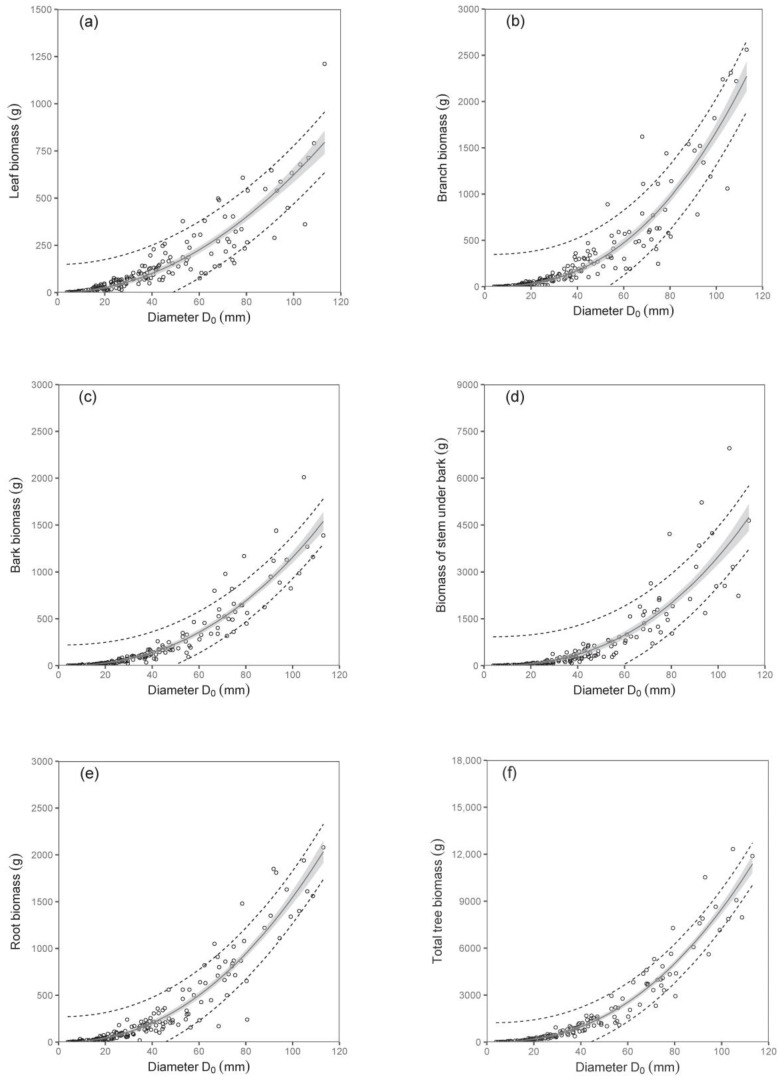
Relations between diameter D_0_ and leaf biomass (diagram (**a**)), branch biomass (**b**), bark biomass (**c**), biomass of stem under bark (**d**), root biomass (**e**), and total tree biomass (**f**) for silver birch (characteristics of allometric models are shown in Table 3 and Table 4). Gray areas indicate confidence intervals (95%), and dashed lines denote ranges of prediction intervals (95%).

**Figure 3 plants-12-01607-f003:**
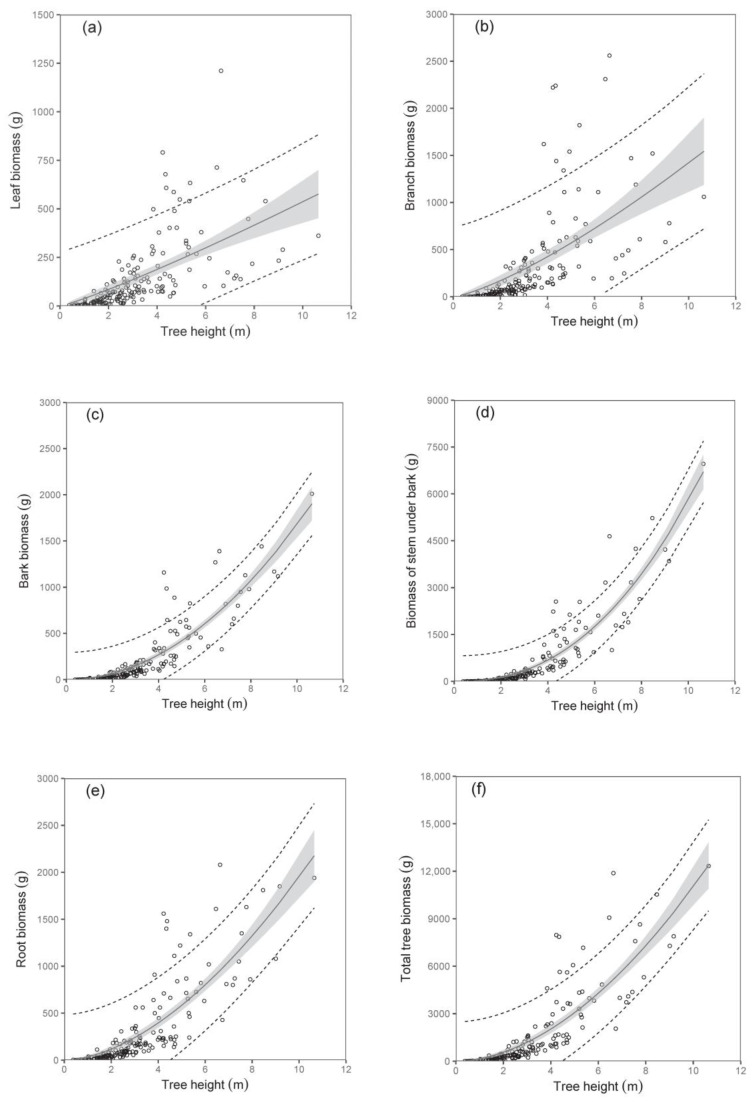
Relations between tree height and leaf biomass (diagram (**a**)), branch biomass (**b**), bark biomass (**c**), biomass of stem under bark (**d**), root biomass (**e**), and total tree biomass (**f**) for silver birch (characteristics of allometric models are shown in Table 3 and Table 4). Gray areas indicate confidence intervals (95%), and dashed lines denote ranges of prediction intervals (95%).

**Figure 4 plants-12-01607-f004:**
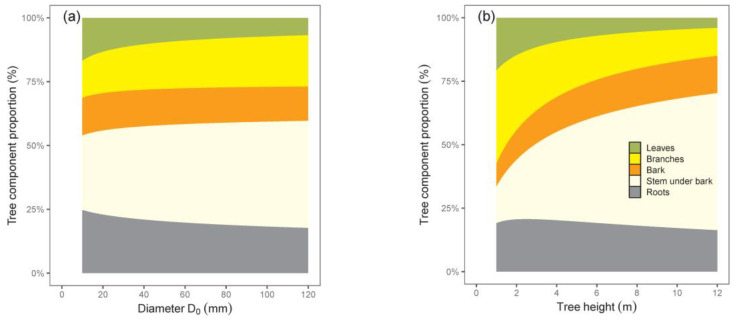
Contributions of tree components, specifically leaves, branches, stem under bark, bark, and roots to total tree biomass with regard to diameter D_0_ (diagram (**a**)) or tree height (**b**) for silver birch.

**Figure 5 plants-12-01607-f005:**
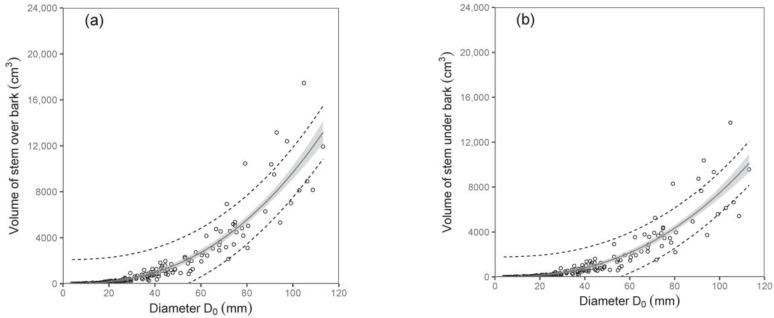
Relations between diameter D_0_ (diagrams (**a**,**b**)) or tree height (**c**,**d**) and stem volume under bark (right diagrams) and over bark (left diagrams) for silver birch (characteristics of allometric models are shown in Table 5). Gray areas indicate confidence intervals (95%), and dashed lines denote ranges of prediction intervals (95%).

**Figure 6 plants-12-01607-f006:**
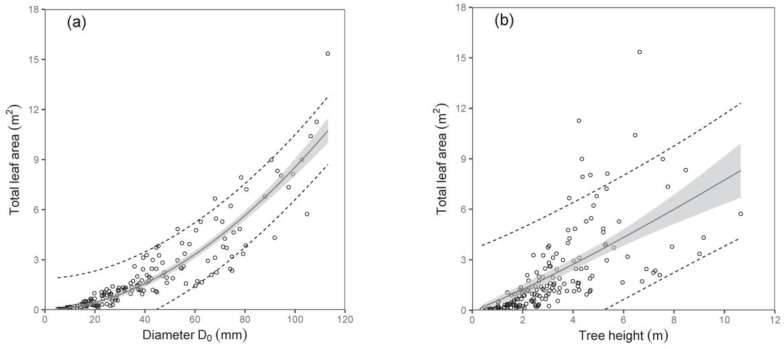
Relations between diameter D_0_ (diagram (**a**)) or tree height (**b**) and total tree leaf area for silver birch (characteristics of allometric models are shown in Table 6). Gray areas indicate confidence intervals (95%), and dashed lines denote ranges of prediction intervals (95%).

**Figure 7 plants-12-01607-f007:**
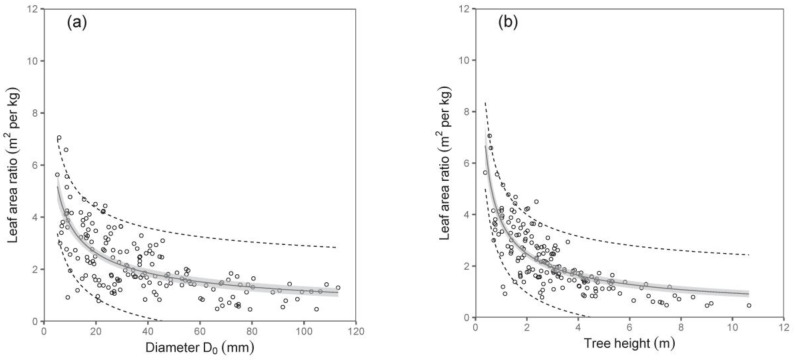
Relations between diameter D_0_ (diagram (**a**)) or tree height (**b**) and leaf area ratio (LAR) for silver birch (characteristics of allometric models are shown in Table 6). Gray areas indicate confidence intervals (95%), and dashed lines denote ranges of prediction intervals (95%).

**Figure 8 plants-12-01607-f008:**
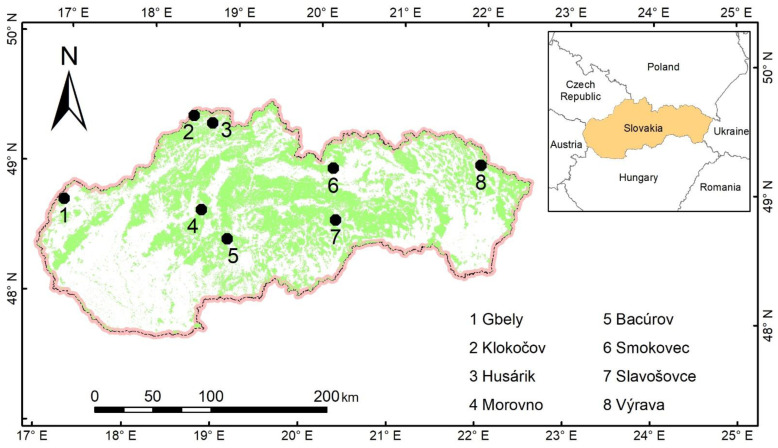
Localization of sampled sites. The legend shows names of the sites next to their numeric codes. The green background illustrates the forest area in the Western Carpathians, Slovakia.

**Table 1 plants-12-01607-t001:** Descriptive statistics of all sampled trees of silver birch in the Western Carpathians, Slovakia (n = 180).

Tree Variable	Unit	Mean	S.D.	Minimum	Maximum	25th Percentile	75th Percentile
Diameter D_0_	(mm)	38.49	25.69	3.90	113.10	18.80	53.05
Diameter DBH	(mm)	19.94	15.61	1.40	61.15	7.58	29.50
Height	(m)	3.12	1.93	0.38	10.65	1.75	4.18
Leaf biomass	(g)	137.95	181.25	1.35	1210.94	21.03	166.06
Branch biomass	(g)	304.44	479.39	0.15	2560.00	25.65	334.56
Stem biomass under bark	(g)	594.61	1054.35	0.75	6959.55	36.00	636.65
Bark biomass	(g)	213.20	333.78	0.55	2010.00	21.64	211.57
Root biomass	(g)	301.74	439.89	0.90	2080.00	35.52	306.84
Aboveground biomass	(g)	1253.98	1947.32	2.80	10,391.13	115.48	1282.21
Total biomass	(g)	1559.48	2380.80	3.90	12,331.13	157.88	1622.57

**Table 2 plants-12-01607-t002:** Models describing relationships between stem base diameter (D_0_) and tree height (H), as well as between diameter at breast height (DBH) and stem base diameter (D_0_) for silver birch. Abbreviations b_0_, b_1_, b_2_—regression coefficients, S.E.—standard error, *p*—*p* value, R^2^—coefficient of determination, MSE—mean square error, RMSE—root mean square error, AIC—Akaike information criterion, BIC—Bayesian information criterion.

Related Variables	Unit	Equation	b_0_	S.E.	*p*	b_1_	S.E.	*p*	b_2_	S.E.	*p*	R^2^	RMSE	AIC	BIC
H vs. D_0_	(m)	(1)	−11.136	20.250	0.583	10.251	1.214	<0.001	0.046	0.014	<0.001	0.78	0.91	476	489
DBH vs. D_0_	(mm)	(2)	−6.627	0.733	<0.001	0.611	0.015	<0.001	-	-	-	0.92	4.39	881	890

**Table 3 plants-12-01607-t003:** Models describing relationships between stem base diameter (D_0_) or tree height (H) and biomass of tree components, i.e., leaves, branches, stem under bark, bark, and roots for silver birch. The abbreviations are explained in the caption of Table 2.

Related Variables	Unit	Equation	b_0_	S.E.	*p*	b_1_	S.E.	*p*	R^2^	RMSE	AIC	BIC
leaf biomass vs. D_0_	(g)	(3)	0.068	0.027	0.014	1.982	0.091	<0.001	0.83	74.93	2036	2046
leaf biomass vs. H	(g)	(4)	38.580	8.490	<0.001	1.143	0.128	<0.001	0.39	140.99	2260	2270
branch biomass vs. D_0_	(g)	(3)	0.019	0.009	0.036	2.475	0.105	<0.001	0.87	175.36	2311	2321
branch biomass vs. H	(g)	(4)	68.074	18.445	<0.001	1.320	0.152	<0.001	0.39	373.87	2576	2585
stem biomass vs. D_0_	(g)	(3)	0.037	0.022	0.100	2.490	0.135	<0.001	0.80	465.23	2683	2692
stem biomass vs. H	(g)	(4)	26.698	4.830	<0.001	2.336	0.089	<0.001	0.85	412.08	2640	2649
bark biomass vs. D_0_	(g)	(3)	0.028	0.011	0.014	2.308	0.090	<0.001	0.89	111.10	2139	2149
bark biomass vs. H	(g)	(4)	16.845	2.935	<0.001	1.998	0.088	<0.001	0.80	147.96	2239	2248
root biomass vs. D_0_	(g)	(3)	0.059	0.020	0.004	2.210	0.077	<0.001	0.90	136.59	2236	2246
root biomass vs. H	(g)	(4)	35.455	6.939	<0.001	1.741	0.102	<0.001	0.69	243.98	2440	2450

**Table 4 plants-12-01607-t004:** Models describing relationships between stem base diameter (D_0_) or tree height (H) and aboveground or total tree biomass for silver birch. The abbreviations are explained in the caption of Table 2.

Related Variables	Unit	Equation	b_0_	S.E.	*p*	b_1_	S.E.	*p*	R^2^	RMSE	AIC	BIC
abvg. biomass vs. D_0_	(g)	(3)	0.112	0.037	0.003	2.397	0.074	<0.001	0.93	525.08	2664	2674
abvg. biomass vs. H	(g)	(4)	121.200	24.283	<0.001	1.875	0.103	<0.001	0.73	1021.95	2895	2904
total biomass vs. D_0_	(g)	(3)	0.160	0.051	0.002	2.363	0.070	<0.001	0.93	618.76	2690	2699
total biomass vs. H	(g)	(4)	157.320	31.163	<0.001	1.846	0.102	<0.001	0.73	1249.33	2930	2939

**Table 5 plants-12-01607-t005:** Models describing relationships between stem base diameter (D_0_) or tree height (H) and stem volume under bark (V_sub_) or stem volume over bark (V_sob_) for silver birch. The abbreviations are explained in the caption of Table 2.

Related Variables	Unit	Equation	b_0_	S.E.	*p*	b_1_	S.E.	*p*	R^2^	RMSE	AIC	BIC
V_Sub_ vs. D_0_	(cm^3^)	(3)	0.100	0.053	0.062	2.438	0.119	<0.001	0.83	901.93	2917	2927
V_Sub_ vs. H	(cm^3^)	(4)	74.752	13.717	<0.001	2.194	0.091	<0.001	0.82	938.44	2931	2941
V_Sob_ vs. D_0_	(cm^3^)	(3)	0.101	0.050	0.045	2.490	0.110	<0.001	0.86	1057.39	2973	2983
V_Sob_ vs. H	(cm^3^)	(4)	96.732	18.207	<0.001	2.184	0.094	<0.001	0.81	1225.76	3026	3035

**Table 6 plants-12-01607-t006:** Models describing relationships between stem base diameter at base (D_0_) or tree height (H) and total tree leaf area (LA), as well as leaf area ratio (LAR) for silver birch. The abbreviations are explained in the caption of Table 2.

Related Variables	Unit	Equation	b_0_	S.E.	*p*	b_1_	S.E.	*p*	R^2^	RMSE	AIC	BIC
LA vs. D_0_	(m^2^)	(3)	0.002	<0.001	0.004	1.851	0.078	<0.001	0.85	0.95	485	495
LA vs. H	(m^2^)	(4)	0.564	0.112	<0.001	1.137	0.116	<0.001	0.43	1.84	717	726
LAR vs. D_0_	(m^2^ kg^−1^)	(3)	11.744	1.317	<0.001	−0.500	0.038	<0.001	0.51	0.87	437	447
LAR vs. H	(m^2^ kg^−1^)	(4)	3.744	0.105	<0.001	−0.597	0.035	<0.001	0.62	0.76	394	403

**Table 7 plants-12-01607-t007:** Main characteristics of sampling sites used for developing tree biomass and foliage area allometric relations in the Western Carpathians, Slovakia.

Site Code	Site Name	Altitude	Latitude	Longitude	Geomorphological
		(m a.s.l.)	(° N)	(° E)	Subprovince
1	Gbely	171	48.7289	17.1025	Borská Lowland
2	Kolokočov	692	49.4621	18.5413	Zapadné Beskydy Mts.
3	Husárik	750	49.4143	18.7692	Javorníky Mts.
4	Morovno	619	48.7368	18.7188	Vtáčnik Hills
5	Bacúrov	449	48.5271	19.0423	Štiavnické Hills
6	Smokovec	890	49.1291	20.2310	High Tatra Mts.
7	Slavošovce	640	48.7301	20.2928	Slovenské rudohorie Mts.
8	Výrava	505	49.2068	21.9747	Nízke Beskydy Mts.

## Data Availability

Data are available through the corresponding author upon reasonable request for academic purposes.

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
