# Peer review of "Tree Biomass and Leaf Area Allometric Relations for Betula pendula Roth Based on Samplings in the Western Carpathians"

_plants, 2023, doi:10.3390/plants12081607_

Round 1
Reviewer 1 Report
Paper is based on very good data set.
There are many papers about Betula spp. Biomass allocation, that miss in this paper, for example:
Johansson, T. 1999a. Biomass equations for determining fractions of pendula and pubescent birches growing on abandoned farmland and some practical implications. Biomass and Bioenergy. 16: 223 – 238.
Martiník, A., Knott. R., Krejza J., Černý J. 2018. Biomass utilization of Betula pendula Roth stands regenerated in the region of allochthonous Picea abies (L.) dieback. Silva Fenica, 52 (5), 15 p.
Uri, V., Varik, M., Aosaar, J., Kanal, A., Kukumägi, M., Lőhmus, K. 2012. Biomass production and carbon sequestration in a fertile silver birch (Betula pendula Roth) forest chronosequence. Forest Ecology and Management 267:117–126.
Incorporation of these papers in discussion can improve quality of this paper.
Please correct the Latin name of birch in all paper - Roth no Roth. .
Author Response
We would like to express thanks to both reviewers for their useful comments on the manuscript. The comments were considered and nearly all incorporated into the text. The changes helped to improve the manuscript.
R1:
There are many papers about Betula spp. Biomass allocation, that miss in this paper, for example:
Johansson, T. 1999a. Biomass equations for determining fractions of pendula and pubescent birches growing on abandoned farmland and some practical implications. Biomass and Bioenergy. 16: 223 – 238.
Martiník, A., Knott. R., Krejza J., Černý J. 2018. Biomass utilization of Betula pendula Roth stands regenerated in the region of allochthonous Picea abies (L.) dieback. Silva Fenica, 52 (5), 15 p.
Uri, V., Varik, M., Aosaar, J., Kanal, A., Kukumägi, M., Lőhmus, K. 2012. Biomass production and carbon sequestration in a fertile silver birch (Betula pendula Roth) forest chronosequence. Forest Ecology and Management 267:117–126.
Incorporation of these papers in discussion can improve quality of this paper.
Authors:
We agree, the citations of the papers were implemented in the Introduction section
R1:
Please correct the Latin name of birch in all paper - Roth no Roth. .
Authors:
Yes, we corrected the Latin name in the entire text.
Reviewer 2 Report
The manuscript "Tree Biomass and Leaf Area Allometric Relations for Betula Pendula Roth. based on Samplings in the Western Carpathians" by Konôpka et al. presents the allometric models of total biomass in young silver birch (Betula pendula Roth.) trees and the main components, i.e. leaves, branches, stem under bark, bark and roots. The subject fits the scope of the journal.
Although the manuscript presents the results of a regional study, the topic is very current and is elaborated comprehensively. I appreciate that the study also deals with root biomass, which is often neglected. I found the paper interesting and generally well‐written, easy to follow. However, the study lacks validation of the derived models. Therefore I recommend publishing the paper after major revision.
Detailed comments:
Model validation – please supplement following analyses:
- - Cross-validation - if no explicit validation data set is available, cross-validation should be used to predict the model fit to a hypothetical validation set. Mean error of cross-validation then expresses how accurately a predictive model will perform in practice.
- - Testing of biomass additivity, i.e. the equalness of the sum of component biomass estimates with the directly estimated aboveground or total biomass.
Minor remarks:
L 302 – 311 it would be clearer to provide information about the sample sites in the form of a table
- Fig. 8 improve the north marking (no overlapping between arrow and “N”); please insert a map of Europe with marked location of Slovakia
- L 363 – 364 root biomass and total tree biomass is missing in the list of tree components
- L 383-385 add the definition of basic wood density
Author Response
We would like to express thanks to both reviewers for their useful comments on the manuscript. The comments were considered and nearly all incorporated into the text. The changes helped to improve the manuscript.
R2:
Model validation – please supplement following analyses:
- - Cross-validation - if no explicit validation data set is available, cross-validation should be used to predict the model fit to a hypothetical validation set. Mean error of cross-validation then expresses how accurately a predictive model will perform in practice.
- - Testing of biomass additivity, i.e. the equalness of the sum of component biomass estimates with the directly estimated aboveground or total biomass.
Authors:
We have completed the cross-validation and commented on the procedure in the Materials and Methods section. The results of the cross-validation have been included into a new Table A1, which is included in Appendix A.
Thank you for your comment on the additivity problem. In fact, after reviewing the available literature focused on allometric relations we might state that most of the authors (over 80%) did not implement the principle of additivity in their models. Actually, our previous experience proved that there were more-less negligible differences in biomass quantities expressed by the relationship for the entire tree and the sum of values coming from models of specific tree components. We are convinced that the question of additivity implementation in biomass models is more theoretical than practical issue. Moreover, there are two other reasons why we prefer not to implement the additivity approaches in the models:
- the final (additive) model has a complicated form that is rather unfriendly for users,
- the random errors produced from the additive model do not have a constant variance.
R2:
L 302 – 311 it would be clearer to provide information about the sample sites in the form of a table
Authors:
Good point. Thus, we have prepared an extra table (Table 7) showing characteristics of the sites.
R2:
Fig. 8 improve the north marking (no overlapping between arrow and “N”); please insert a map of Europe with marked location of Slovakia
Authors:
We agree, the Fig. 8 was modified according to the comments from the reviewer.
R2:
- L 363 – 364 root biomass and total tree biomass is missing in the list of tree components
Authors:
Yes, they were added.
R2:
L 383-385 add the definition of basic wood density
Authors:
Good idea. The definition was added.
Round 2
Reviewer 2 Report
The authors improved the original manuscript and did a perfect work. Most of my comments have been incorporated. I appreciate the newly added Table A1, which gives the readers a clear idea of model accuracy. I have only a few minor comments on the current form of the manuscript, which will be easy to incorporate (see below). My recommendation is to publish the manuscript after incorporating these minor comments.
Specific comments:
Biomass additivity problem: in my previous review I did not ask for developing new models with implemented additivity. My recommendation was only to calculate the error when estimating the total above-ground or total biomass using the single model in comparison with the error if we calculate these values as the sum of the individual tree components. My experience is that single model performs better.
l. 423 please add following explanation „yi - observed value“
Table A1 The units must be given for Mean absolute error as well as for MSE and RMSE. Regarding MSE and RMSE, I believe that listing only RMSE would be sufficient. I would recommend also stating the Relative mean error of cross-validation in %, which gives the readers a clear idea about the accuracy of the estimation by the given model.
Author Response
Specific comments:
Biomass additivity problem: in my previous review I did not ask for developing new models with implemented additivity. My recommendation was only to calculate the error when estimating the total above-ground or total biomass using the single model in comparison with the error if we calculate these values as the sum of the individual tree components. My experience is that single model performs better.
We fully agree with the reviewer. Therefore, we calculated the RMSE when estimating the total aboveground and total biomass using the single model in comparison with the RMSE if we calculate these values as the sum of the individual tree components. Description of the statistical procedure is incorporated in Material and Methods section and introduced in this way:
- 389 “In addition to developed biomass models for tree components, the aboveground biomass and the total tree biomass were also obtained by summing the model predictions of tree components to explore the consistency of biomass additivity.”
- 437 “To evaluate additivity of tree biomass components, we calculated the RMSE when estimating the total aboveground and total biomass using the single model in comparison with the RMSE calculated as the sum of the individual tree components.”
The analyses showed slightly lower values of RMSE favouring developed single regression models for estimating total aboveground biomass and total biomass. These findings were included in the Results section as follows:
- 124 “The RMSE obtained for the estimated total aboveground and total biomass with diam-eter D0 as predictor variable was 527.66 g and 618.76 g, respectively. Slightly higher values of RMSE were obtained for estimated total aboveground (RMSE = 527.82 g) and total biomass (RMSE = 618.96 g) calculated as the sum of the individual tree compo-nents. The RMSE obtained for the estimated total aboveground and total biomass with tree height H as predictor variable was 1,027.36 g and 1,249.33 g, respectively. RMSE values for the estimated total aboveground and total biomass calculated as the sum of the individual tree components reached 1,037.94 g and 1,260.11 g, respectively.”
- 423 please add following explanation „yi - observed value“
OK, we added the missing notation related to the formula.
Table A1 The units must be given for Mean absolute error as well as for MSE and RMSE. Regarding MSE and RMSE, I believe that listing only RMSE would be sufficient. I would recommend also stating the Relative mean error of cross-validation in %, which gives the readers a clear idea about the accuracy of the estimation by the given model.
OK, we agree with the reviewer. Thus, we have added the units in the Tables 2 – 6 (in fact, it was missing there as well). Then, we did not repeat the unit description anymore in the Table A1 but we referred about the units in the table caption to the previous tables. As for MSE, we deleted the column showing its values since this was redundant with RMSE. A new column was added showing relative RMSE (RMSE%).
